# Effect of Transglutaminase Treatment on the Structure and Sensory Properties of Rice- or Soy-Based Hybrid Sausages

**DOI:** 10.3390/foods12234226

**Published:** 2023-11-23

**Authors:** Mirian dos Santos, Wanessa Oliveira Ribeiro, Jamille de Sousa Monteiro, Bibiana Alves dos Santos, Paulo Cezar Bastianello Campagnol, Marise Aparecida Rodrigues Pollonio

**Affiliations:** 1School of Food Engineering, Universidade Estadual de Campinas (Unicamp), Cidade Universitária Zeferino Vaz, Campinas 13083-862, SP, Brazil; m192335@dac.unicamp.br (M.d.S.); wanessaor@hotmail.com (W.O.R.); j203680@dac.unicamp.br (J.d.S.M.); 2Department of Food Science and Technology (DTCA), Universidade Federal de Santa Maria, Av. Roraima 1000, Camobi, Santa Maria 97105-900, RS, Brazil; bialvessantos@gmail.com (B.A.d.S.); paulocampagnol@gmail.com (P.C.B.C.)

**Keywords:** meat reduction, vegetable proteins, texture, microstructure, projective mapping

## Abstract

Partial substitution of meat with non-protein sources in hybrid meat products generally leads to a decrease in texture attributes and, consequently, in sensory acceptance. In this study, we investigated the effects of transglutaminase (TG) at two concentrations (0.25% and 0.5%) on the physicochemical, textural, and sensory properties of hybrid sausages formulated with concentrated soy or rice proteins. TG caused a reduction in the heat treatment yield of hybrid sausages, particularly those made with rice protein. pH and color parameters were marginally affected by TG addition. Texture parameters increased substantially with TG, although escalating the TG level from 0.25% to 0.5% did not result in a proportional improvement in texture parameters; in fact, for rice-based hybrid sausages, no difference was achieved for all attributes, while only cohesiveness and chewiness were improved for soy-based ones. TG enhanced the sensory attributes of soy-based hybrid sausages to a level comparable to control meat emulsion, as evidenced by ordinate preference score and projective mapping. Our findings suggest that TG is a viable strategy for enhancing texture and sensory parameters in hybrid sausages, particularly for plant proteins that exhibit greater compatibility with the meat matrix.

## 1. Introduction

Hybrid meat products are elaborating by partially substituting meat with more sustainable protein sources. These products address both sustainability and nutritional concerns, since reduced-meat products help to mitigate the environmental impact associated with livestock farming and may offer an enhanced nutritional profile, as they combine essential nutrients found in meat, such as: high-quality proteins; key minerals like heme iron, zinc; as well as B-complex vitamins with dietary fibers, bioactive compounds; and an array of additional minerals and vitamins sourced from plants [1,2]. Various plant proteins, including soy, pea, rice, and sunflower, can be utilized in the formulation of hybrid meat products, nevertheless, the effects among these proteins can vary considerably [3]. 

Despite the advantages of hybrid meat products, there are some challenges that must be taken into account. For example, production costs could be high depending on the ingredients used and additional process steps. Regulatory issues present another significant challenge, as hybrid meat products do not easily align with existing meat or plant-based food categories. 

Additionally, developing a sensory profile that appeals to both traditional meat consumers and flexitarians can be an obstacle. The texture parameters substantially decreased in hybrid meat products elaborating with concentrated plant proteins due to the different gelation behaviors of plant and meat proteins during cooking [4,5] In meat products, myosin is the main protein responsible for the texture due to its fibrous structure that facilitates the interaction between adjacent protein chains resulting in a high gelation behavior. In contrast, plant proteins showed well-organized structures characterized by high levels of hydrophobic associations. Consequently, elevated temperatures are necessary to unfold these protein chains to achieve suitable gelation. Additionally, it is relevant to note that concentrated proteins often undergo rigorous extraction processes that can potentially affect their functional properties [6].

Soy is a first option for hybrid and analog meat products since it has an excellent functional and nutritional profile with high bioavailability. Soy proteins primarily comprise salt-soluble globulins like B-conglycinin (7S) and glycinin (11S). These globulins have great water-holding, foaming, and emulsifying capacities [7]. Nutritionally, soy proteins contain all the essential amino acids and have a high Digestible Indispensable Amino Acid Score (DIAAS) of 91 [8]. In a previous study, we demonstrated that soy protein exhibits excellent functionality with the meat matrix, making it a suitable candidate for use in hybrid emulsified products [3]. 

Despite its superior techno-functional properties, soy protein is considered allergenic, and therefore its use may limit the consumption of soy-based hybrid meat products. On the other hand, rice protein is considered to be hypoallergenic. It is also tasteless and colorless. Most of the protein in rice is comprised of an alkali-soluble glutelin which, despite its poor emulsifying properties, shows suitable gelling abilities due to its high content of disulfide bonds [9]. Concerning nutritional content, rice protein has an unbalanced amino acid profile, with a DIAAS of 47 and lysine as its limiting amino acid [8]. 

Various strategies can enhance protein gelation, including the utilization of high-pressure and ultrasonic technologies, as well as enzymatic treatment with transglutaminase (TG) [10,11,12]. Among these approaches, enzymatic treatment stands out as a more cost-effective and easier method, suitable for both small-scale and large-scale production. TG enhances texture by facilitating protein cross-linking, amine incorporation, and deamidation, thereby strengthening protein gels and increasing water-holding capacity [13]. In meat products, TG acts as a texture enhancer, leading to a firmer and more cohesive texture, and can also improve emulsion stability and decrease the cooking loss [12]. Hybrid meat products can also benefit from TG activity, mainly for improving the texture-related parameters. 

Given that rice and soy have differing protein compositions—with rice primarily composed of glutelins and soy of globulins—it is crucial to understand the role of TG in optimizing texture in hybrid sausages made with either soy or rice proteins, which could be useful in developing more appealing hybrid meat products. Therefore, the objective of this study is to evaluate the effects of two different concentrations of transglutaminase in hybrid sausages formulated with soy and rice proteins. 

## 2. Materials and Methods

### 2.1. Ingredients

An outside flat cut of bovine meat was obtained from a local market in Campinas, SP, Brazil. The meat was trimmed to remove fat and visible connective tissue before being ground to a 5 mm particle size using a meat grinder (Model STB 22 CAF, Rio Claro, SP, Brazil). The ground meat (76.18 ± 0.11 moisture, 23.5% protein, 3.00% fat, pH 5.90 ± 0.03) was vacuum-sealed and stored at −20 ± 2 °C for two weeks until further use. Concentrated soy protein (80% crude protein, 3.1% fat, pH 7.53—Seara Foods, Osasco, SP, Brazil), concentrated rice protein (74% crude protein, 3.3% fat, pH 6.50—Growth Supplements, Tijucas, SC, Brazil), vegetal fat AlConf P45XST (Saturated fat: 68%; monounsaturated fat: 17%; polyunsaturated fat: 13%—Cargill, Campinas, SP, Brazil), transglutaminase Activa TG-N-SF (activity of 78–126 U/g, Ajinomoto Brazil, Limeira, SP, Brazil), sodium erythorbate and sodium nitrite (Kerry, Campinas, SP, Brazil), and sausage seasoning (NewMax, Americana, SP, Brazil) were donated by the respective companies. Canola oil Liza (Cargill, Brazil) was purchased from the local market.

### 2.2. Experimental Design and Hybrid Sausage Preparation

Seven treatments were formulated, as described in Table 1, across three independent batches: red meat sausage (FC), hybrid sausages with soy (FS) and rice (FR) proteins, and hybrid sausages treated with 0.25% (FST1 and FRT1) and 0.50% TG (FST2 and FRT2). A vegetable fat blend was prepared by thoroughly mixing melted vegetable fat and room-temperature canola oil in a 1:1 *w/w* ratio. This blend was rapidly cooled to −18 °C and subsequently stored at 4 °C. Both soy and rice protein concentrates were hydrated with cold water (4 °C) to achieve a 20% protein content, based on their initial crude protein content, just before use.

Sausage formulations were prepared in a Ninja Auto IQ Pro food processor (Model BL492BZ-30, Hai Xin Technology, Shuzhen, China). First, immediately before processing, the vacuum-packed meat was thawed in a water bath until it reached approximately −4 °C. Thus, meat (−4 °C), hydrated vegetable protein (4 °C), NaCl, sausage condiment, sodium nitrite, transglutaminase, and a portion of ice were well-homogenized. Sodium erythorbate, the vegetable fat blend, and the remaining ice were then incorporated and mixed until the temperature reached 14 °C. Plastic tubes were filled with approximately 40 g of the raw sausage and then refrigerated at 6 °C for 3 h before cooking. Heat treatment was conducted in a water bath (Model MA 093, Marconi, Piracicaba, SP, Brazil), starting from room temperature to 80 °C for approximately 50–55 min. After cooking, the samples were immediately cooled in an ice water bath and stored at 4 °C for 48 h until subsequent analyses.

### 2.3. Heat Treatment Yield, pH, and Instrumental Color Analyses

Heat treatment yield (HTY) was evaluated by comparing the initial and final weights of the samples before and after the heat treatment process, and the exudate released from the samples was removed with a paper towel. The HTY was expressed in percentage, and the analysis was performed in three repetitions per batch/treatment. The pH was measured directly in the samples at room temperature using a FiveGo F20 pH meter (Mettler Toledo, Greifensee, Switzerland) with four repetitions per batch/treatment. Instrumental color was assessed using a CM-5 spectrophotometer (Konica Minolta, Tokyo, Japan) operating with a D65 illuminant, a 10° observation angle, SCE mode, and the CIELab color system. Three repetitions per batch/treatment were performed. Whiteness (W) was determined using the following formula: $100-\sqrt{\;{{\left(100-L\right)}^{2}+a}^{2}+b^{2}}$ [14].

### 2.4. Evaluation of Pressed Juice

Pressed juice was evaluated following a methodology adapted from a previous study [15]. The sample was cut into a cubic shape with 0.5 cm edges and weighed between two sheets of qualitative filter paper (Ø 12.5 cm). The sample was then compressed by a cylindrical probe (Ø 50 mm) for 30 s at 78.45 N using a TA-XT2i Texturometer (Texture Technologies Corp., Hamilton, MA, USA), equipped with a 25 Kg load cell and a trigger force of 5 g. After compression, the sample was removed, and the filter paper was reweighed. The pressed juice yield was expressed as a percentage, and the analysis was conducted with four repetitions per batch/treatment.

### 2.5. Texture Profile Analysis (TPA)

Texture Profile Analysis (TPA) was conducted using a TA-XT2i Texture Analyzer (Texture Technologies Corp., Hamilton, MA, USA) equipped with a 25 Kg load cell and a 5 g trigger force. Six samples per batch/treatment, measuring 1.5 cm in height and 2.0 cm in diameter, were axially compressed in two consecutive cycles with a 5-s interval and 50% compression. A cylindrical probe (Ø 50 mm) was used at a constant speed of 2 mm/s. Data were analyzed for hardness (N), springiness (mm), cohesiveness (dimensionless), and chewiness (N·mm).

### 2.6. Microstructure Analysis 

The sausage samples, each measuring 0.5 cm^2^ in area and 0.1 cm in height, were initially frozen at −60 °C using the CL200-86V Ultra Freezer (Coldlab, Piracicaba, SP, Brazil). They were then freeze-dried for 48 h in a Super Modulyo freeze-dryer (Edwards, Irvine, CA, USA), maintaining a condenser temperature of −55 °C and a pressure of 4.0 × 10^−2^ mbar. The microstructure of the freeze-dried samples was evaluated using a TM 4000 tabletop scanning electron microscope (SEM) from Hitachi Technologies, Tokyo, Japan. The SEM settings were as follows: an accelerating voltage of 10 kV (in mode 4), an emission current of 52,000 nA, and a vacuum level of 50 (Chg-up Red. L). The Backscattered Electron (BSE) detector was utilized in shadow 2 mode. Representative images were captured at 150× magnification with a scale bar corresponding to 300 µm.

### 2.7. Sensory Analysis

Sensory analysis was approved by the Ethics in Research Committee at Unicamp, SP, Brazil, under CAAE number 65068922.2.0000.5404. A total of 30 consumers, aged between 18 and 61 years (57% women and 43% men), participated in this study. Consumers evaluated five samples, each coded with a three-digit identifier. These samples were distributed to consumers simultaneously, following a balanced design. Initially, consumers were asked to rank the samples based on their preferences in the Ordination Preference Test. Subsequently, a Projective Mapping analysis was employed following a previous methodology [16], with minor modifications: consumers were instructed to place the samples on a blank A4 sheet of paper (210 × 297 mm), arranging them based on their perceived similarities or differences. Two closely placed samples were interpreted as similar, while those placed farther apart were considered different. Participants were informed that there were no right or wrong answers. After arranging the samples, participants were asked to describe between three and six attributes that characterized each sample or group of samples. 

### 2.8. Statistical Analysis 

Data were analyzed using a General Linear Model (GLM) with a 95% confidence level (*p* ≤ 0.05), followed by post-hoc Tukey’s test (*p* ≤ 0.05) in IBM SPSS Statistics 20 software. In the Ordination Preference Test, the relative positioning of the samples, ranging from least to most preferred, was converted into preference scores (1 for least preferred to 5 for most preferred). This data was then assessed using the non-parametric Friedman’s test, followed by the post-hoc Fisher’s Least Significant Difference (LSD) test. The distribution of consumer preferences was further analyzed through Internal Preference Mapping in XLSTAT 2022.3.2 software (Addinsoft, New York, NY, USA). Projective Mapping was evaluated using the STATIS method in XLSTAT, relying on the X and Y coordinates of the samples as positioned on a two-dimensional plane on an A4 sheet of paper. For descriptive analysis, words were assessed according to two criteria: (1) derivatives were reduced to their root form and (2) synonyms or terms referring to levels of intensity were grouped together. Multiple Factor Analysis (MFA) was conducted in XLSTAT 2022.3.2 software, utilizing the frequency of each sensory descriptor as a variable.

## 3. Results and Discussion

### 3.1. Heat Treatment Yield (HTY), Pressed Juice, pH and Color Parameters of Hybrid Sausages Elaborated with Soy or Rice Proteins and Transglutaminase 

The HTY, displayed in Table 2, varied from 71.7% to 96.20%. The treatments with rice protein showed the lowest HTY, and the soy ones showed the highest. Extrinsic factors such as pH and ionic strength affect protein solubilities and, consequently, their emulsifying, foaming, water-holding capacity, and gelation properties [17]. As previously reported, rice proteins are primarily composed of alkali-soluble glutelin that has high hydrophobicity and virtually no solubility at the pH presented in the sausage treatments, so its interaction with water is significantly impaired. A previous study demonstrated that rice glutelins had poor solubilities at a wide range of pH (2 to 10), and their interfacial properties are favored in strong acid conditions (at pH 2) [18]. The better HTY in soy-based hybrid sausages could be correlated to the incorporation of 2% NaCl and pH near 6.50, which contributed to better solubilities of both soy globulins and myofibrillar meat proteins. Salt levels at intermediate ionic strength (0.1–0.6 N) neutralize the surface charges of proteins at the oil-water interface, which reduces electrostatic repulsion and facilitates an increased rate of protein adsorption [19]. These factors contribute to enhanced emulsion stability, which in turn results in a higher HTY, as evidenced in soy-hybrid sausages. 

The FST2 treatment shows a minor reduction in HTY. In contrast, the rice-based hybrid sausages showed a significant decline in HTY with increasing levels of transglutaminase (TG) in the formulation. Specifically, the yield diminished from 82.57% in the FR treatment to 71.57% in the FRT2 treatment. The cross-links between rice proteins promoted by TG possibly led to the formation of insoluble protein aggregates that diminish the proteins’ ability to interact with water, thereby leading to water release [20,21]. According to a previous study, the increase in cross-linkages between protein chains caused a narrow space among them, leading to more water release during the heat process [22].

Pressed juice (PJ) expresses the water-holding capacity (WHC) of the food matrix. The WHC is an indicator of a protein gel’s ability to bind water, mainly stabilized through electrostatic interactions and hydrogen bonds. Additionally, WHC is linked to the gel’s capacity to stabilize water molecules via capillary effects, which are closely related to pore size [23,24]. Interestingly, TG caused less PJ loss, as shown in Table 2, for both soy and rice treatments. The TG leads to the formation of covalent bonds between the plant and meat protein molecules, increasing protein network strengthening that allows water to be retained within the meat matrix under external influence, such as a compressive force [25]. Similarly, other studies showed higher WHC with TG addition in pork meat gels [26], meat added with hemp protein and flaxseed flour [26], and soy-based analog burgers [27]. 

The pH values of the treatments were influenced by both plant proteins and the TG incorporation (Table 2). Soy-based sausages showed higher pH (6.44 to 6.51) than those with rice (6.09 to 6.11) and FC treatment (6.06). pH values far from the isoelectric points (pI) of proteins increase their solubility and functional properties [17]. In meat, myofibrillar proteins showed a pI close to a pH of 5.0 [28], 7S and 11S soy globulins have a pI near a pH of 4.0 [7], and for rice glutelin, the pI is approximately at pH 5 [18]. Thus, the higher HTY presented by soy-based sausages could be attributed to the great functional properties of soy protein and the pI drifts for both soy and myofibrillar proteins. TG also caused a slight increase in pH values of the treatments that could be related to the phosphates aggregated as vehicle agents in the formulation of Activa TG-S-NF.

In meat products, color is subject to variations based on the ingredients used and the processing steps employed. In general, the primary determinants influencing this parameter are the levels of myoglobin, fat, water, and non-meat ingredients [29]. Color coordinates were slightly affected by TG incorporation in soy-based hybrid sausages (Table 2 and Figure 1). For rice-based ones, lightness (L*) decreased, and redness (a*) increased with TG addition in rice-based sausages. An elevated water loss during heat treatment led to a concentration of both nitrohemocrome and the intrinsic pigments present in the rice protein. This concentration contributed to an increase in a* and a corresponding decrease in L* in the rice treatments. Other studies showed a decreased value of L* with TG incorporation; according to them, TG contributes to the denaturation of proteins and, consequently, to the lower L* values [27,30]. The soy-based hybrid sausages did not show an expressive water loss and color parameters were marginally affected in treatment FST2 compared to FS. The images of the sausage treatments corroborated the color attributes (Figure 1).

### 3.2. Texture Parameters and Microstructure of Hybrid Sausages Elaborated with Soy or Rice Proteins and Transglutaminase 

All texture parameters increased with TG incorporation in both soy and rice hybrid sausages (Figure 2). The hardness was improved considerably with TG, and it was more pronounced in sausages with soy protein compared to those containing rice protein. This result may have correlated to the differences in protein composition between soy and rice proteins. Soy is primarily composed of globulins, which likely exhibited increased solubility due to the salt content and pH levels in the soy hybrid sausages. Better protein solubility would increase the number of available active sites for enzymatic cross-linking. This phenomenon may account for the observed increase in hardness in soy-based sausages treated with transglutaminase (TG), as seen in the FST treatments, when compared to rice-based hybrid sausages. A prior study demonstrated that TG action was higher when protein solubility increased [31]. As previously discussed, at a pH near 6.00, rice glutelins showed poor solubility, and they are stabilized mainly for disulfide bonds and hydrophobic interactions [32]. Disulfide bonds closed the protein structure and diminished the cross-linking reaction promoted by TG due to an insufficient exposure of the lysine and glutamine residues of proteins [33]. A previous study assessed the effects of transglutaminase (TG) on barley, soy, and wheat proteins. The authors demonstrated that TG was more effective in modifying soy protein compared to barley, which, similar to rice, primarily consists of glutelins as its main protein fractions [34]. 

Regarding springiness and cohesiveness, higher values were observed for the soy-based hybrid sausage (FS) as compared to the rice-based one (FR) that supports the superior compatibility of soy protein with meat matrix. TG incorporation significantly elevated these parameters, most notably in rice-based hybrid sausages, compared to the baseline treatments without TG. It is possible that the large rice protein aggregates strengthen the protein network, causing more resistance to deformation and a better recovery to the original conformation after compression. Since chewiness is a product of the other parameters, it followed their tendencies. Several studies have shown a significant improvement in the texture parameters with TG treatment in meat products [25,30,35,36], soy-based products [37,38], rice protein gels [39], and analog meat products [27,40].

It was interesting that most texture parameters plateaued when TG concentrations increased from 0.25% to 0.5%. This suggests that TG enzymatic activity likely reaches a saturation point near the 0.25% concentration, corroborating findings from a preceding study that showed a plateau behavior on textural properties as increased TG level in myofibrillar protein gel [41] 

In summary, while the TG effect was more discernible in soy-based hybrid sausages for hardness, it significantly improved springiness and cohesiveness in both soy and rice formulations. These findings align well with the main objective of this study, especially since the TG-treated soy-based sausages (FST1 and FST2) closely matched the hardness levels of traditional meat sausages. 

In our study, notable differences were observed in the topographic characteristics of the hybrid sausages, as evidenced in Figure 3. Sausage formulated with soy protein (FS) and the control treatment (FC) presented a uniform and cohesive topography characterized by smooth areas distributed across the surface. On the contrary, sausage elaborated with rice protein (FR) exhibited a porous and rough microstructure that is visible on the sausage surface (Figure 1). 

The addition of transglutaminase (TG) had a pivotal impact on the microstructure characteristics of hybrid sausages, mainly for rice-hybrid sausages. TG led to a denser protein matrix, characterized by reduced pore size and increased surface density in rice-hybrid sausages, which was corroborated by the sausage images (Figure 1). This result is in alignment with our earlier findings regarding the textural parameters. The dense protein matrix is likely a result of the enzymatic cross-linking activity of TG, which correlates with the increase in texture-related parameters. Other studies demonstrated similar behavior on the microstructure of food matrix treated with TG [25,37,38].

### 3.3. Sensory Analysis of Hybrid Sausages Elaborated with Soy or Rice Proteins and Transglutaminase 

The sensory analysis was realized by excluding the treatments containing a 0.5% TG since the texture parameters for these samples were largely indistinguishable from those containing a 0.25% TG level. The results from the Ordination Preference Test, illustrated in Figure 4, show that FST1 and FC treatments were the most preferred by the consumers. Several studies demonstrated that the effect of TG contributes to better sensory acceptance of foods [27,30,36]. Despite the significant achievements in improving texture parameters in soy, and mostly in rice hybrid sausages, the sensory characteristics of rice hybrid sausages were not improved. One possible explanation for this may be the significant water loss during cooking, which probably resulted in a drier texture for FRT1. 

Projective mapping analysis, presented in Figure 5a, supports the ordination preference test outcomes. The first component of the analysis explained 57% of the variation among treatments, effectively separating FC, FS, and FST1 from FR and FRT1 samples. Notably, FST1 was more closely aligned with FC than it was with FS, indicating its higher sensory preference, as can be seen in Figure 5b, as evidenced by the red circle on the internal preference mapping graph. The second component accounted for an additional 18% of the variation and could separate FR from FRT1. FRT1 showed higher water loss during cooking, which likely led to the concentration of salt and other flavor components that adversely affected the sample’s flavor and juiciness, a finding corroborated by attributes like salted, aftertaste, bad flavor, and brittle that were attributed to this treatment in the multiple factor analysis (Figure 5b). Otherwise, FC, FS, and FST1 were positively correlated with desirable sensory attributes, such as juicy, tasty, good aroma, and good appearance. 

## 4. Conclusions

Hybrid meat products may offer a more sustainable alternative to conventional meat products, offering a pathway to reduce red meat consumption. However, partial meat substitution by plant proteins may diminish textural qualities and reduce the sensory acceptability of these products. Our study provides evidence that the enzymatic treatment with transglutaminase can effectively improve the texture and sensory acceptability of hybrid sausages elaborated with plant proteins. This outcome was especially evident in soy-based hybrid sausage, where TG not only improved textural parameters but also led to higher sensory acceptability that was not different from traditional sausage. However, the intrinsic compatibility of the selected plant protein and the meat matrix needs to be considered since, for rice-based hybrid sausages, TG significantly improved textural attributes but failed to enhance sensory quality. In conclusion, transglutaminase offers a promising strategy for improving the textural and sensory attributes of hybrid meat products, although the degree of success may vary depending on the type of plant protein used. 

## Figures and Tables

**Figure 1 foods-12-04226-f001:**
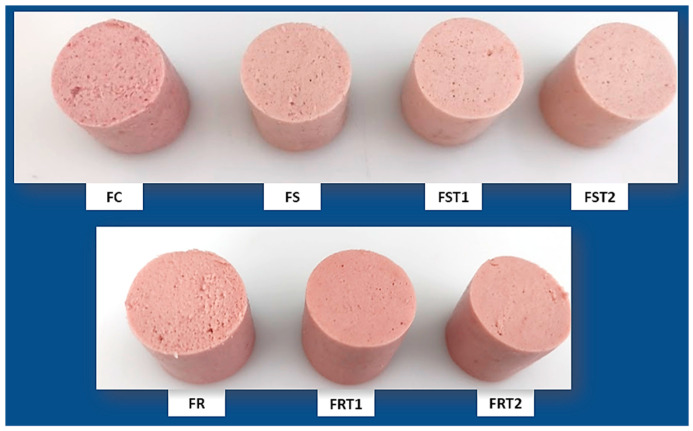
Images of hybrid sausages elaborated with soy or rice proteins and transglutaminase. FC: red meat sausage; FS, FST1, and FST2: soy-based hybrid sausages with 0%, 0.25%, and 0.50% of transglutaminase, respectively; FR, FRT1, and FRT2: rice-based hybrid sausages with 0%, 0.25%, and 0.50% of transglutaminase, respectively.

**Figure 2 foods-12-04226-f002:**
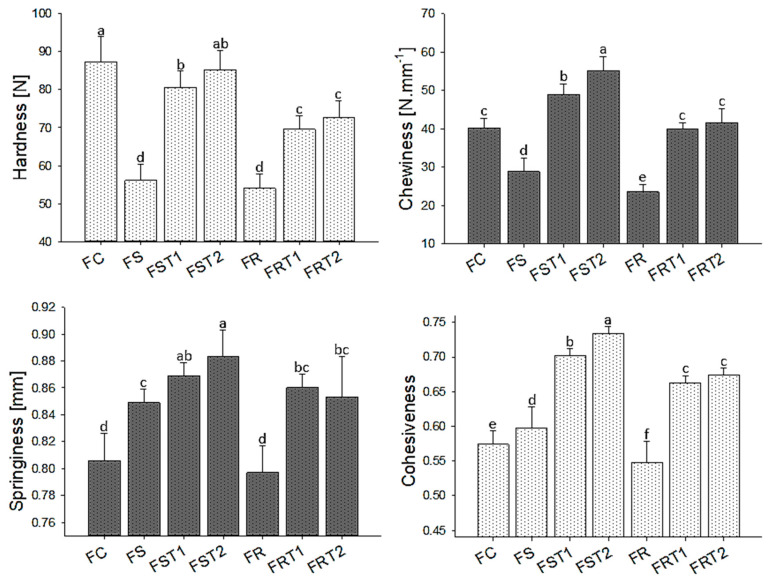
Mean values for texture parameters of hybrid sausages elaborated with soy or rice proteins and transglutaminase. FC: red meat sausage; FS, FST1, and FST2: soy-based hybrid sausages with 0%, 0.25%, and 0.50% of transglutaminase, respectively; FR, FRT1, and FRT2: rice-based hybrid sausages with 0%, 0.25%, and 0.50% of transglutaminase, respectively. Mean values with different letters on the columns of each parameter differ from each other (*p* ≤ 0.05) according to the post hoc Tukey’s test.

**Figure 3 foods-12-04226-f003:**
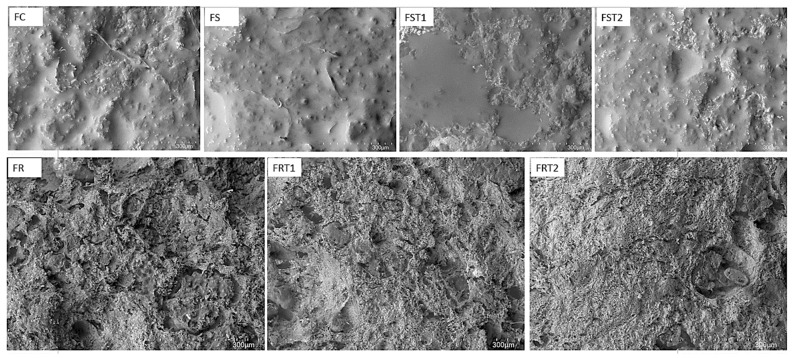
Microstructure images (at a magnification of 150×) of hybrid meat emulsions elaborated with soy or rice proteins and transglutaminase. FC: red meat sausage; FS, FST1, and FST2: soy-based hybrid sausages with 0%, 0.25%, and 0.50% of transglutaminase, respectively; FR, FRT1, and FRT2: rice-based hybrid sausages with 0%, 0.25%, and 0.50% of transglutaminase, respectively.

**Figure 4 foods-12-04226-f004:**
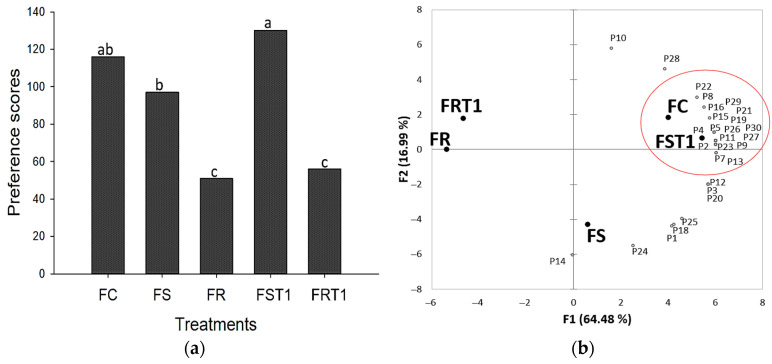
(**a**) Scores of Ordination Preference Test and (**b**) Internal Preference Mapping Distribution for hybrid sausages elaborated with soy or rice proteins and transglutaminase. FC: red meat sausage; FS and FST1: soybased hybrid sausages with 0% and 0.25% of transglutaminase, respectively; FR and FRT1: rice-based hybrid sausages with 0% and 0.25% of transglutaminase, respectively. Mean values with different letters on the columns of each parameter differ from each other (*p* ≤ 0.05) according to the post hoc LSD test.

**Figure 5 foods-12-04226-f005:**
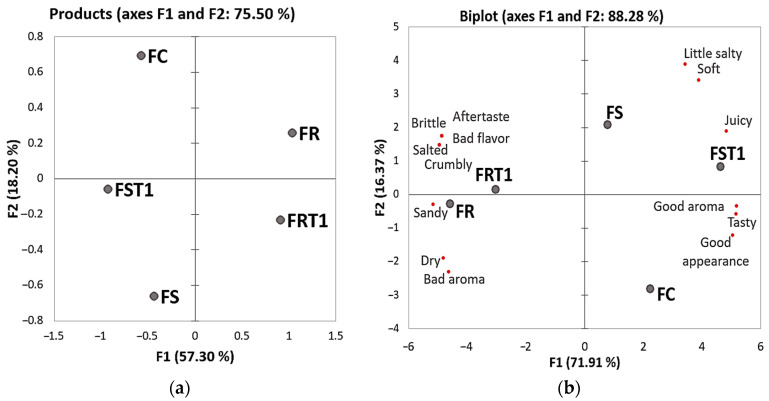
(**a**) Projective Mapping and (**b**) Multiple Factor Analysis for hybrid sausages elaborated with soy or rice proteins and transglutaminase. FC: red meat sausage; FS and FST1: soy-based hybrid sausages with 0% and 0.25% of transglutaminase, respectively; FR and FRT1: rice-based hybrid sausages with 0% and 0.25% of transglutaminase, respectively.

**Table 1 foods-12-04226-t001:** Formulations (g/100 g) of hybrid sausages elaborated with soy or rice proteins and transglutaminase enzyme.

Ingredients	Treatments						
	FC	FS	FST1	FST2	FR	FRT1	FRT2
Lean red meat	65.00	32.50	32.50	32.50	32.50	32.50	32.50
Hydrated soy protein	-	32.50	32.50	32.50	-	-	-
Hydrated rice protein	-	-	-	-	32.50	32.50	32.50
Transglutaminase TG-N-SF	-	-	0.25	0.50	-	0.25	0.50
Water	12.94	12.94	12.69	12.44	12.94	12.69	12.44

All treatments contained 1.3% sodium chloride, 1.0% sausage condiment (70% NaCl), 20% vegetal fat blend, 0.015% sodium nitrite, and 0.05% sodium erythorbate. FC: red meat sausage; FS, FST1, and FST2: soy-based hybrid sausages with 0%, 0.25%, and 0.50% of transglutaminase, respectively; FR, FRT1, and FRT2: rice-based hybrid sausages with 0%, 0.25%, and 0.50% of transglutaminase, respectively.

**Table 2 foods-12-04226-t002:** Mean values (±standard deviation) of heat treatment yield (HTY), pressed juice, pH, and color parameters of hybrid meat emulsions elaborated with soy or rice proteins and transglutaminase.

Parameters	Treatments						
	FC	FS	FST1	FST2	FR	FRT1	FRT2
Heat treatment yield [%]	87.91 (1.12) ^B^	95.83 (0.61) ^A^	96.20 (0.37) ^A^	95.37 (0.36) ^B^	82.50 (1.32) ^C^	76.41 (1.38) ^D^	71.57 (1.16) ^E^
Pressed juice [%]	15.34 (1.58) ^D^	22.14 (0.95) ^B^	15.64 (1.55) ^D^	16.62 (1.75) ^D^	27.56 (1.02) ^A^	23.00 (2.13) ^B^	20.24 (1.17) ^C^
pH	6.06 (0.01) ^F^	6.44 (0.01) ^C^	6.48 (0.02) ^B^	6.51 (0.02) ^A^	6.09 (0.02) ^E^	6.09 (0.02) ^E^	6.11 (0.01) ^D^
Lightness [L*]	61.34 (0.54) ^D^	64.91 (0.20) ^A^	64.75 (0.44) ^AB^	64.24 (0.69) ^B^	63.21 (0.38) ^C^	59.67 (0.53) ^E^	59.57 (0.33) ^E^
Redness [a*]	11.22 (0.39) ^A^	8.47 (0.22) ^D^	8.53 (0.15) ^D^	8.51 (0.30) ^D^	9.39 (0.11) ^C^	10.38 (0.21) ^B^	10.42 (0.13) ^B^
Yellowness [b*]	12.93 (0.44) ^C^	14.95 (0.25) ^B^	15.09 (0.26) ^B^	15.46 (0.14) ^A^	15.11 (0.12) ^B^	15.45 (0.23) ^A^	15.27 (0.23) ^AB^
Whiteness [W]	57.72 (0.73) ^D^	60.93 (0.29) ^A^	60.72 (0.37) ^A^	60.12 (0.30) ^B^	59.14 (0.40) ^C^	55.58 (0.60) ^E^	55.54 (0.40) ^E^

Mean values with different letters in the same row differ from each other (*p* ≤ 0.05) according to the post hoc Tukey’s test. FC: red meat sausage; FS, FST1, and FST2: soy-based hybrid sausages with 0%, 0.25%, and 0.50% of transglutaminase, respectively; FR, FRT1, and FRT2: rice-based hybrid sausages with 0%, 0.25%, and 0.50% of transglutaminase, respectively.

## Data Availability

The data presented in this study are available on request from the corresponding author.

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
