# Peer review of "Effect of Transglutaminase Treatment on the Structure and Sensory Properties of Rice- or Soy-Based Hybrid Sausages"

_foods, 2023, doi:10.3390/foods12234226_

Round 1

Reviewer 1 Report

Comments and Suggestions for Authors

The manuscript is presenting evaluate the effects of two different additive concentrations of transglutaminase (TG 0.25 and 0.5%) in hybrid sausages formulated with soy and rice proteins on the selected their quality properties.

The manuscript is interesting and fits in the scope of the Foods journal.

The abstract of the reviewed paper has informative and fully covered the content of the manuscript. The length of the manuscript is appropriate in relation to the content.

In the reviewer’s opinion the objective of the study is correctly formulated and sufficiently described. The experimental design and methods are appropriate for the purposes of the study, and investigations has been conducted in an ethically acceptable manner.

The article has been prepared in accordance with the instructions for authors. English is understandable. This article requires some minor clarifications prior in Foods journal.

After reading through the manuscript, I found several issues that should be addressed by the authors:

1.     Based on what criteria was the TG addition determined at 0.25 and 0.5%?

2.     Why grounded meat was frozen before its used to sausages production? (line 103-104), and next:

-      what method was used to freeze the sample?

-      what was the freezing time?

-      what method was used to thaw the samples?

-      what was the thawing time?

3.     Line 131 and next: in the reviewer’s opinion more appropriate term would be "heat treatment" – the cooking process is provided at a temp. 100°C

4.     What was the storage time of the samples after heat treatment? (line 131-132)

5.     Line 147 – consistently it should be: “heat treatment”.

6.     Subsection 2.6. “Microstructure analysis” – What was the process of freeze-drying the produced sausages?

7.     Subsection 2.7. “Sensory analysis” – Please explain whether the research carried out included a sensory evaluation or a consumer evaluation? The description shows that a consumer evaluation has been carried out.

8.     Subsection 3.1. “Physicochemical characterization ……”, Table 2. – the specified parameters are not physicochemical, but physical parameters, except the pH values! Moreover, instead of the incorrect term "luminosity" it should be: "lightness" or "photometric lightness" and …. “heat treatment yield” instead “ cooking yield”.

 The manuscript should after minor revision.

Reviewer 2 Report

Comments and Suggestions for Authors

The work focuses on the use of the enzyme transglutaminase (TG) in the preparation of hybrid sausages with soy or rice, to improve the rheological characteristics of the hybrid products.

Line 2,3,4: Title

I suggest changing the title: “Effect of transglutamase treatment on the structure and sensory properties of rice- or soy-based hybrid sausages”

Line 34: Keywords: more central and specific;

Line 44: Introduction

I think the introduction is too generic. It would be appropriate to discuss more about the choice to produce hybrid meat products and give more information on the costs of such production, as well as the problems related to the implementation of this production. In this section it would also be useful to explain the choice of vegetable source: why rice and why soy?

Line 99: Materials and Methods

L 103: Section.2.2: Experimental design and hybrid sausage preparation

Minced meat (76.18 ± 0.11 moisture, 23.5% protein, pH 5.90 ± 0.03). Why did the authors choose not to report the percentage of lipids?

L 107-108: AlConf P45XST vegetable fat (Saturated fats: 68%; monounsaturated fats: 17%; polyunsaturated fats: 13%.

I suggest reporting the concentrations of the fatty acids of each class, especially the saturated ones.

Line 133-134: Table 1: please report the abbreviations in the legend;

 L 143: Check the percentage of water added FTR1 and FTR2.

 Line 187: Section 2.7. Sensory analysis

If the analysis is a consumer test the number of consumers seems to be too low (30 subjects). Otherwise change consumers with panelists  

In general: Materials and methods: I suggest evaluating lipid and protein oxidation in order to strengthen the quality aspect of cooked soy-based hybrid rice or sausage.

Line 219: Section: Results and discussions

Section 3.1.: Physicochemical characterization of hybrid sausages elaborated with soy or rice proteins

Reporting only pH, color etc. is too simplistic. Chemical characterization would be very useful for the nutritional evaluation of the studied products.

L 241-245: being an explanation, I believe it should be inserted in the initial part of the section where the results of the rice cooking yield are reported (L224).

 Line 401: Section: Conclusions

“Hybrid meat products can offer nutritional balance”. This statement was not supported by the results as there was no adequate chemical-physical characterization.
